# The Use of Respiratory Effort Improves an ECG-Based Deep Learning Algorithm to Assess Sleep-Disordered Breathing

**DOI:** 10.3390/diagnostics13132146

**Published:** 2023-06-23

**Authors:** Jiali Xie, Pedro Fonseca, Johannes P. van Dijk, Xi Long, Sebastiaan Overeem

**Affiliations:** 1Biomedical Diagnostics Lab, Department of Electrical Engineering, Eindhoven University of Technology, 5612 AZ Eindhoven, The Netherlands; pedro.fonseca@philips.com (P.F.); dijkh@kempenhaeghe.nl (J.P.v.D.); s.overeem@tue.nl (S.O.); 2Philips Research, High Tech Campus, 5656 AE Eindhoven, The Netherlands; 3Sleep Medicine Center Kempenhaeghe, 5591 VE Heeze, The Netherlands; 4Department of Orthodontics, Ulm University, 89081 Ulm, Germany

**Keywords:** electrocardiogram, ECG-derived respiration, respiratory effort, sleep-disordered breathing, apnea, recurrent neural network

## Abstract

Background: Sleep apnea is a prevalent sleep-disordered breathing (SDB) condition that affects a large population worldwide. Research has demonstrated the potential of using electrocardiographic (ECG) signals (heart rate and ECG-derived respiration, EDR) to detect SDB. However, EDR may be a suboptimal replacement for respiration signals. Methods: We evaluated a previously described ECG-based deep learning algorithm in an independent dataset including 198 patients and compared performance for SDB event detection using thoracic respiratory effort versus EDR. We also evaluated the algorithm in terms of apnea-hypopnea index (AHI) estimation performance, and SDB severity classification based on the estimated AHI. Results: Using respiratory effort instead of EDR, we achieved an improved performance in SDB event detection (F1 score = 0.708), AHI estimation (Spearman’s correlation = 0.922), and SDB severity classification (Cohen’s kappa of 0.62 was obtained based on AHI). Conclusion: Respiratory effort is superior to EDR to assess SDB. Using respiratory effort and ECG, the previously described algorithm achieves good performance in a new dataset from an independent laboratory confirming its adequacy for this task.

## 1. Introduction

Sleep-disordered breathing (SDB) forms a highly prevalent group of sleep disorders, characterized by repeated cessation (apnea) or reduction (hypopnea) of respiratory airflow during sleep [1]. Sleep apnea can be categorized as obstructive apnea when caused by partial or complete obstruction of the upper airway, central apnea when there is a reduced or absent respiratory drive from the brain stem, and mixed apnea when there is a combination of the two [2]. SDB is linked to several relevant clinical outcomes, including cardiovascular disease; neur×obehavioral impairments, such as sleepiness, fatigue, depressed mood, and impaired memory and concentration; and metabolic dysfunction [2,3]. It is estimated that approximately 13% of men and 6% of women aged 30–70 years have moderate-to-severe SDB [4], with a considerable portion remaining undiagnosed [5]. The gold standard for diagnosing SDB remains polysomnography (PSG), but this technique has several limitations, such as the requirement for many and difficult to set up sensors, high cost, and limited availability [6]. The utilization of home polysomnography has mitigated some of these limitations; however, it remains a relatively complex and costly methodology [7]. Consequently, it may not be ideal for longer-term monitoring in the pre-diagnostic phase or for post-treatment evaluation of the effectiveness of SDB therapy. Therefore, a more convenient and cost-effective screening method is needed.

In 2000, the Computer in Cardiology Challenge released a task to classify one-minute sleep epochs as either presenting or not apneic events, based solely on the electrocardiographic (ECG) signal from a dataset containing 70 overnight recordings [8,9]. The publicly available dataset used for this task was called the “Apnea-ECG database”. This challenge triggered an important increase in research in this area, and since then, various methods have been developed based on ECG signals, some employing ECG-derived respiration (EDR), with many achieving promising results. For instance, Tripathy [10] used a kernel extreme learning machine classifier on features extracted from the intrinsic band functions of both heart rate variability (HRV or RR interval) and EDR signals and obtained a sensitivity and a specificity for the detection of sleep apnea epochs of 78.02% and 74.64%, respectively. Yang et al. [11] proposed a deep learning algorithm named 1-D squeeze-and-excitation residual group network for OSA detection using HRV and EDR and achieved a sensitivity of 87.6% and a specificity of 91.9%. Zarei and Asl [12] applied a GentleBoost classifier on features from HRV and EDR and reached a sensitivity of 91.52%, a specificity of 94.36%, and an accuracy of 93.26%. Although these very high detection performances could indicate that the problem has been adequately resolved, it has been suggested that results obtained exclusively on the Apnea-ECG database should be considered only as a starting point for apnea detection research. With an insufficient variety of sleep pathologies and SDB severity, models trained exclusively on this dataset are likely overfitted and might not generalize well to new data that are more representative of population or clinical cohorts [13].

More recently, Olsen et al. [14] proposed a method based on heart rate and EDR using recurrent neural network (RNN) for detecting SDB events (including hypopnea, obstructive apnea, central apnea, and mixed apnea) on a large dataset including almost 10,000 recordings of various population cohorts. His model reached an overall sensitivity of 70.9%, a precision of 73.4%, and an F1 score of 72.1%. However, there are studies arguing that EDR might not be the best signal to measure respiratory activity for SDB detection. Varon et al. [15] found that information is reduced when transferring from respiration to heart rate during apnea and, as a result, the use of EDR as a surrogate of respiration might in fact reduce the performance for SDB detection, in particular when detecting hypopneas [16]. Sadr et al. [17] and Deviaene et al. [18] found a decreased apnea detection performance using EDR compared to thoracic respiratory effort (RE) measured with respiratory inductance plethysmography (RIP). Although the argument of developing a model based solely on a single ECG sensor is attractive, RE or surrogates thereof are among the most easily obtained signals using a variety of unobtrusive methods, such as in-bed pressure sensor arrays [19], chest-worn accelerometers [20], and even wrist-worn reflective photoplethysmography [21]. Notably, these sensing methods are notoriously easy to use, relatively comfortable, and thus, suitable for deployment at home in longer-term monitoring scenarios. Additionally, these sensors could also obtain the cardiac activity extracted by these studies from ECG [22,23,24]. Therefore, combining ECG and RE would most likely be advantageous for more accurate SDB monitoring.

The RNN-based algorithm proposed by Olsen et al. [14] was chosen for this study as it shows promising results on large population cohorts, and because the EDR can be easily replaced by RE to allow for a direct comparison. In addition, RNN can automatically extract features from the signals, unlike in the other two studies [17,18] that compared RE with EDR using manually engineered features extracted from the two signals. If these features were originally developed for one sensing modality (i.e., RE), it is plausible that they fail to fully capture the same information with the other (EDR), without necessarily meaning that EDR is inadequate for the SDB detection task. By training an RNN directly on either signal, we can let if fully exploit any available information on each modality.

In this study, we aimed to validate the deep RNN algorithm proposed by Olsen et al. [14] in an independent clinical dataset and compare the performance obtained using EDR versus using RE measured with RIP belts. The performance was evaluated in terms of SDB event detection, apnea-hypopnea index (AHI) estimation, and SDB severity classification.

## 2. Materials and Methods

### 2.1. Dataset

The dataset used in this study is a subset of the SOMNIA database [25]. We retrieved 198 subjects who were over the age of 18 and were not being treated with continuous airway pressure therapy, without any selection requirements for body mass index (BMI), sex, or AHI. All subjects in our dataset underwent routine PSG monitoring at the Kempenhaeghe Center for Sleep Medicine, Heeze, the Netherlands, between June 2017 and November 2017. The dataset was manually annotated based on the full PSG by sleep technicians using the 2015 AASM guidelines. In total, there were 198 subjects included with 121 male and 77 female, an average age of 50.1 ± 14.8 years, an average BMI of 27.2 ± 4.8 kg/m^2^, and an average AHI of 18.0 ± 18.4 events/hour. Among all signals from PSG, only ECG and RE signals were used in this study. ECG signal was measured by electrodes from Kendall (Ashbourne, Ireland) and RE signal was measured by RIP belts from Sleepsense (Elgin, IL, USA).

### 2.2. Signal Pre-Processing and Segmentation

R-peaks were first detected from the ECG signals using a previously developed algorithm, which is based on nonlinear transformation and a simple peak-finding strategy [26]. Subsequently, a post-processing algorithm based on the intersection of tangents fitted to the slopes of the R-waves was applied to achieve precise localization of the QRS complexes and to eliminate artifacts [27]. To address the presence of ectopic beats, the algorithm proposed by Mateo and Laguna [28] was employed, which identifies abnormal beats by imposing restrictions on the allowable acceleration or deceleration of the heart rate. The resulting RR intervals were linearly interpolated and resampled at 4 Hz to obtain an RR time series. Periods containing artifacts and ectopic beats were marked with a value of 0 to exclude them from further analysis. The algorithm considered these as missing values and learned how to handle them without the need of using a masking layer.

The EDR signal was obtained using the phase-space reconstruction method proposed by Janbakhshi and Shamsollahi [29], and it was also resampled at 4 Hz to have the same sample frequency as the RR signal. Regarding the RE signal measured with RIP, for consistency, we also resampled the raw signal to 4 Hz, followed by a high-pass filter with a cut-off frequency of 0.05 Hz applied to eliminate low-frequency noise [30]. During the resampling step, high-frequency (>2 Hz) noise was also removed.

Afterwards, we segmented all signals into 5-minute segments with a 2-minute overlap, resulting in a data shape of 1200 × 1 per segment. Each segment of signal was normalized using “soft” min-max normalization, where the minimum value was set to the 5th percentile and the maximum value to the 95th percentile. Segments at the beginning and end of the recording, where the lights were turned on, were excluded from the analysis. For each segment, two different combinations of signals (either RR and EDR, or RR and RE) were used to form an input shape of 1200 × 2 for model fitting and for inference. Each combination was achieved by directly concatenating segments from the two different signals. The network was trained to autonomously learn the underlying relationship between the two inputs.

### 2.3. Deep Learning Architecture

We endeavored to replicate the RNN-based algorithm described by in [14], henceforth referred to as “Olsen’s algorithm”. The architecture is detailed in Table 1 and Figure 1 and is composed of three blocks. The first two blocks are employed for feature extraction and share an identical structure. Each block comprises two layers of bidirectional gated recurrent units (GRU) to capture temporal dependencies in the time series, a batch normalization layer to mitigate the internal covariate shift, a max-pool layer to reduce the output dimensionality, an activation layer with rectified linear unit (ReLU) activation, and a dropout layer to prevent overfitting. The last block consists of a dense-connected layer using the ReLU activation function and a dense-connected layer with the sigmoid activation function to generate the output, which corresponds to a value between 0 and 1, indicating the probability of an SDB event at 1 Hz.

During training, we used the Adam optimizer with a learning rate of 0.001 and a weight decay of 0.0001. The model was initialized using a robust initialization scheme [31], and a batch size of 128 was selected. Furthermore, in contrast to Olsen’s algorithm, we included kernel regularization to counteract overfitting and employed sample weighting (label 0:1, label 1:10) to account for the imbalance in our dataset. The training was terminated when the evaluation loss did not improve for more than 20 training epochs to further avoid overfitting. No underfitting was observed, as both the training and the validation performance achieved high accuracy on the classification task at hand.

The objective of the algorithm was to detect all SDB events including hypopneas, obstructive apneas, central apneas, and mixed apneas. Each target vector corresponded to a 5-minute segment with a resolution of 1 s, resulting in a vector length of 300 s. Each second was assigned a value of 1 if it belonged to an SDB event and 0 if not. During the training phase, the model was trained using input segments to generate their corresponding target vectors. The patterns for distinguishing between normal and abnormal breathing were learned by the model. Similar to the training process of Olsen’s algorithm, no additional techniques were employed. In this work, we trained two distinct models for SDB event detection. The first model used RR and EDR as inputs (RR + EDR), while the second model used RR and RE (RR + RE).

### 2.4. Performance Evaluation

The dataset was split into training, validation, and testing sets using a stratified random split approach. First, all subjects were categorized into four SDB severity groups based on their AHI (normal: AHI < 5, mild: 5 < AHI < 15, moderate: 15 < AHI < 30, severe: AHI > 30) [32]. Next, approximately 25% of the subjects from each severity group were randomly selected for the testing. Similarly, around 75% and 25% of the remaining subjects from each group were included in the training and the validation set, respectively. This resulted in 114, 35, and 49 subjects separated between the training, validation, and testing sets, respectively. The training set was used to fit the model, while the validation set was used for early stopping and deriving the decision threshold. The test set was held out exclusively for evaluating the classifier. Table 2 shows the characteristics of the participants and corresponding data used for training, validation, and testing sets after splitting. The imbalance of different classes and the variety of subjects are also shown in this table.

To provide a comprehensive analysis of the performance, we evaluated the algorithm in terms of SDB event detection, AHI estimation, and SDB severity classification. To assess the performance of SDB event detection over the entire night, the segment-based outputs pertaining to the same night were concatenated to generate a continuous output series of a whole night for each subject. Specifically, the central 3-minute interval within each 5-minute segment was concatenated, while the first and the last minute of each segment were excluded from evaluation to ensure adequate context awareness for each detection [14]. Detected SDB events were obtained by combining the consecutive seconds with probabilities exceeding a specified threshold, while periods with probabilities below the threshold were considered as non-SDB.

A true positive (TP) event was defined as the first detected event that overlapped with an annotated SDB event. Events that did not overlap with or were not the first detected event that overlapped with an SDB event were considered as false positive (FP) events. A false negative (FN) event was an SDB event for which no overlap was found any detected event. The threshold used to detect SDB events was selected based on the maximum F1 score over all SDB events on the validation set. Concretely, we applied a threshold from 0.004 to 0.996 with a step of 0.004 on the SDB event detection results from the validation set and computed the corresponding F1 scores. The threshold with the maximum F1 score was selected to compute the results for the testing set. Note that our TP definition was stricter than that of Olsen et al. [14] since we considered only the first overlapped event as TP, whereas they reported all detected events as TPs if they overlapped with any ground truth SDB event. This can lead to an overestimation of the number of TPs and consequent underestimation of FP. The discrepancy between the two definitions can be observed in Figure 2, where a hypothetical scenario is presented as an illustrative example to underscore the contrasting interpretations. In addition, and in contrast with the study by Olsen et al. [14], we did not find any benefit in eliminating short events or combining close events into a single event.

SDB event detection performance was evaluated by calculating the sensitivity, precision, and F1 score per subject in the testing set and after aggregating the detection results from all test subjects. Additionally, the detection rate (sensitivity) of different types of SDB events was computed. When evaluating the results, we only counted SDB events detected during sleep (as scored based on PSG). Event probabilities output by the model during wake periods were set to zero. Using total sleep time, also obtained from the PSG scoring, we computed the AHI (total number of detected SDB per night divided by total sleep time).

For AHI estimation, a Bland–Altman analysis between the estimated AHI (AHI_est_) and reference (AHI_ref_) is presented. AHI_ref_ was derived from the ground truth SDB events annotated from PSG. In addition, scatter plots (i.e., estimated versus reference values), the corresponding Spearman’s correlation coefficient R, and the intraclass correlation coefficient (ICC) [33,34], using two-way random-effects model for measuring the agreement between two scorings, are provided. Because most patients in our cohort did not have a very large AHI value (>30) and the correlation analysis can be very sensitive to just a few large values, we also calculated correlation coefficients for subjects with an AHI less than 30.

Furthermore, we evaluated our method in SDB severity classification, separating between normal (AHI < 5), mild (5 < AHI < 15), moderate (15 < AHI < 30), and severe (AHI > 30), based on the estimated AHI values. Confusion matrix, accuracy, and Cohen’s Kappa were computed based on these canonical thresholds and computed with near boundary double-labeling (NBL) [30,31]. NBL was used to mitigate overly negative evaluations of SDB severity caused by AHI values very close to the SBD severity boundaries. With NBL, when the AHI of a subject (according to the PSG reference) is close to the boundary between two SDB severity classes, the subject is assigned to both possible reference classes. If the subject were classified (with our algorithm) as either of the two possible reference SDB severity classes, it is considered in agreement with the reference. For example, a subject with an AHI value of 5.1 is assigned to both the normal and mild classes, and the classification result will be considered correct if the subject is classified as either normal or mild. In this study, we used the near boundary zones from Pee et al. [35], originally determined by limiting SDB severity disagreement probability of two human PSG scorings on the same dataset to 33%.

Moreover, to gain insights into the impact of age and BMI on the performance, we computed the Spearman’s correlation coefficient (R) between the error of AHI (|AHI_ref_–AHI_est_|) and age and BMI. Additionally, we analyzed the detection rate (sensitivity) in different sleep stages and body positions to explore their association with the event detection performance.

## 3. Results

### 3.1. SDB Event Detection

Table 3 shows the SDB event detection results including the mean and standard deviation (SD) obtained across all test subjects as well as the aggregated results of all events from the test subjects. The results show that the model trained based on RR and RE data outperformed that based on RR and EDR data. Specifically, the model employing RR + RE achieved an average sensitivity of 62.6 ± 26.7%, precision of 50.4 ± 23.6%, and F1 score of 0.529 ± 0.241 as well as a pooled sensitivity of 77.4%, precision of 65.2%, and F1 score of 0.708. In contrast, the model utilizing RR + EDR exhibited lower performance across all metrics. Table 4 presents the pooled SDB event detection rate of different SBD types (hypopneas, obstructive apneas, central apneas, and mixed apneas). Consistent with the results shown in Table 3, using RE instead of EDR improved the detection rate for all SDB types. Notably, the detection rate of hypopneas was much lower than the other three types of SDB, indicating the difficulty of detecting hypopneas from ECG (and RE) signals, consistent with the findings in other studies [14,36,37].

Furthermore, for the model with RR + RE, we found a non-significant Spearman’s correlation of 0.212 (*p* = 0.14) between the error in the number of detected events (i.e., true positives plus false positives) minus the number of actual events according to the ground truth and the reference AHI.

### 3.2. AHI Estimation

Figure 3 shows the Bland–Altman analysis and scatter plots of AHI estimation results in the test set. The 95% limits of agreement between AHI_est_ and AHI_ref_ for RR + RE was −3.16 ± 12.09, lower than that for RR + EDR (−2.82 ± 16.23). Combing RR and RE yielded a higher correlation than using RR and EDR, achieving a correlation coefficient R = 0.922 (*p* < 0.0001) for all test subjects and R = 0.923 (*p* < 0.0001) for test subjects with AHI < 30. Consistently, RR + RE exhibited a higher ICC of 0.939 (95% confidence interval: 0.87–0.97) between AHI_est_ and AHI_ref_ compared to RR + EDR, which had an ICC of 0.894 (95% confidence interval: 0.81–0.94).

### 3.3. SDB Severity Classification

The SDB severity classification results were derived based on AHI. Confusion matrices with and without NBL are presented in Figure 4. Without NBL, using RR + RE improved the classification results with normal, mild, and moderate SDB. After applying NBL, only the classification for the moderate class improved. Table 5 summarizes the performance of SDB severity classification (accuracy and Cohen’s Kappa) with and without NBL. In general, RR + RE demonstrates better performance than RR + EDR. The best performance was achieved by RR + RE with an accuracy of 0.857 and a Kappa of 0.81 with NBL.

### 3.4. Inpacts on Performance from Age, BMI, Seelp Stage, and Body Position (RR + RE)

The correlation coefficient was R = 0.256 (*p* < 0.1) between age and AHI error and R = 0.342 (*p* < 0.05) between BMI and AHI error. Regarding the impact of sleep stage on sensitivity (Appendix A), we observed that sensitivity is lower in N3 (0.448) compared to REM (0.757), N1 (0.784), and N2 (0.772). In terms of body position (Appendix A), sensitivity was found to be lower in the prone position (0.634) compared to the right (0.738), supine (0.804), and left (0.781) position.

## 4. Discussion

We aimed to evaluate a previously published ECG method for detection of sleep disordered breathing in an independent dataset of 198 patients obtained from a clinical database. In addition, we compared the proposed use of EDR versus the use of RIP belt-derived respiratory effort in terms of SDB detection, AHI estimation, and the SDB severity classification performance.

We found that the implementation of the algorithm using RR and EDR on our dataset actually yielded lower performance compared with the previous study. Specifically, the overall F1 score dropped from 0.721 to 0.607, and the correlation decreased from 0.910 to 0.904, along with a decline in other metrics. This outcome can be attributed to several factors. First, our dataset is considerably smaller in size than the dataset used in the study by Olsen et al. [14]. It is possible that their model could have learned more SDB-related information from the larger dataset, leading to better performance. Second, the subject distribution in the two datasets differed. Our stratified, random splitting method ensured that the training and test datasets had similar distribution of subjects in terms of SDB severity. However, the test dataset in the previous study only included 50 “normal” subjects with AHI smaller than 5, which accounted for just 5.2% of the entire test dataset. In contrast, the percentage of subjects with a normal AHI in our testing set was 30.6%. Misdetections seem to occur more on subjects with a smaller AHI than those with a larger AHI.

Regardless, we observed that incorporating some modifications in the model, including sample weighting and kernel regularization, improved performance in our study. We expect that further adjustments may improve generalizability to other, independent datasets, a necessary requirement before such algorithms are implemented in (clinical) practice. In any case, the algorithm’s adequacy to this task remains evident: despite the different nature of the dataset used, where ours comprised a clinical, as opposed to a population cohort in the original study, performance remained relatively high.

In comparison to the studies [10,11,12] described in the Introduction, which demonstrated highly promising performance, the results of this study appear slightly lower. Notably, these studies made use of the Apnea-ECG dataset, which is known to yield superior performance compared to other datasets. This assertion is supported by the findings of Papini et al. [13], who observed a significant decline in performance (sensitivity <55%, false detection rate >40%) when applying algorithms trained on the Apnea-ECG dataset (with sensitivity >85% and false detection rate <20%) to alternative databases encompassing a broader spectrum of apneic events and other sleep disorders. Similarly, Yang et al. [11] reported a decrease in all evaluation metrics when validating their method on a different dataset (from accuracy 90.3%, sensitivity 87.6%, and specificity 91.9% to 75.1%, 61.1%, and 80.8%, respectively). This aspect was one of the considerations that led to our selection of the study conducted by Olsen et al. [14], as it relied on a dataset comprising large population cohorts. In contrast to the Apnea-ECG dataset, the dataset utilized in the present study was collected in real-world conditions in principle providing a more representative sample. Additionally, it should be noted that while the aforementioned studies focused on segment classification with or without SDB events, our results are based on the detection of actual SDB events; this stricter criterion also contributes to the observed differences in performance.

The AHI estimation performance demonstrates great promise despite the inter-subject variability, with an achieved ICC value of 0.939. This level of agreement is comparable to the AHI agreement observed across 10 scorers in PSGs (ICC 0.984; 95% confidence interval: 0.977–0.990) [38] and the AHI agreement across 9 scorers in home sleep testing data (ICC 0.96; 95% confidence interval: 0.93–0.99) [34]. These results highlight the strong concordance between our AHI estimation and established scoring methods, indicating the reliability of the approach.

We found that the using RE in combination with RR signals yielded better performance in all tasks compared to using only ECG-based signals (i.e., RR and EDR). There could be several reasons to explain this improvement. EDR is derived from heartbeat intervals, and thus, it has a low sampling rate (e.g., approximately 1 Hz for an average heart rate of 60 beats per minute). We speculate that this may result in missing high-frequency components of respiration that are informative to detect SDB events. Additionally, it has been demonstrated that in general EDR yields a relatively poor approximation of the respiratory waveform or morphology [15]. Using EDR, it might be hard to capture, for example, increased breathing effort after apnea or hypopnea events.

Although the use of an additional modality next to ECG might sound less appealing since it represents an increase in the number of sensors, it is relevant to remark that this is amongst the more easily acquirable, with unobtrusive methods. In fact, recent developments in physiological sensing suggest that both the RR and the RE time series used as the input can both be accurately measured from bed sensors [19,22], chest-worn accelerometers [20,23,39], or even wrist-worn PPG sensors [21,24]. In contrast, EDR is ECG-specific and likely not easily obtainable with other sensing modalities. On the other hand, including other signals like blood oxygen saturation and airflow, which obviously hold potential to offer valuable additional information, is helpful but remains challenging to obtain reliably when utilizing unobtrusive methods.

Based on our findings, we observed a correlation between AHI error and age as well as BMI. Higher age and BMI were associated with an increased AHI error. However, it is important to note that this correlation might be influenced by AHI severity, as age and BMI are known to be correlated with AHI severity. In Figure 3, we can observe that higher AHIs are associated with a higher AHI error. Nonetheless, further investigations are warranted to establish and validate the influence of age and BMI on AHI error. Regarding the lower sensitivity observed during the N3 stage, it is likely attributed to the limited number of events observed in N3, which poses difficulties for our model to effectively learn from them, resulting in a lower sensitivity. In terms of body position, although the percentages of different types of SDB events vary, we found consistent sensitivities in the right, supine, and left positions. The lower sensitivity observed in the prone position may be attributed to the same reason that causes the lower sensitivity in N3. It is worth emphasizing that further research is required to dive deeper into these relationships, confirm the findings, and provide a comprehensive understanding of the factors influencing AHI error and sensitivity.

It is important to note that the detection rate for hypopneas was lower compared to other SDB types (see Table 4), consistent with the findings by Olsen et al. [14], which is one limitation of this method. After inspecting the results, we found that the correctly detected SDB events were longer than the missed SDB events (Appendix A), and the average length of hypopnea was shorter than that of obstructive apnea and mixed apneas in our dataset (Appendix A). Thus, the event length might be one reason why hypopneas had a lower detection rate, as longer apneas and hypopneas usually correspond to greater changes in HRV [40].

An important limitation of the study is that the sleep–wake information used for providing the SDB event detection results and compute AHI was directly retrieved from the PSG scorings of sleep stages. For practical use, an automatic sleep–wake identification algorithm is required for a fully-fledged SDB assessment system.

Future work could follow multiple directions. We could investigate the reasons underlying the superior performance of RE signals compared to EDR in detecting SDB. Additionally, there is potential in studying and developing methods to improve hypopnea detection rates, for example, by exploring improvements of the deep learning algorithm by adding attention mechanism or transformers. The method should be further validated with larger datasets that contain a wider spectrum of apneic events, additional sleep disorders, and additional somatic comorbidities. Another area of exploration could be the development of a fully automated method for estimating AHI by integrating sleep–wake identification with the SDB detection method.

Finally, to enable actual real-world applications, two important elements should be investigated. First, there is clearly sufficient information in cardiac and respiratory activity measured with ECG and RIP belts to accurately assess SDB severity. However, it is necessary to evaluate whether surrogate measures of cardiac and respiratory effort obtained with other sensors yield a comparable performance. Second, it would be relevant to investigate whether the computational requirements needed for this method would be adequate for implementation in modern architectures, embedded in the monitoring devices, or as part of cloud-based services.

## 5. Conclusions

In this paper, we implemented a previously described state-of-the-art ECG-based deep learning algorithm for SDB detection in an independent clinical dataset and proved the generalizability and robustness of the method. In addition, we investigated the advantage of using RE measured using RIP belts and compared the performance between models using EDR versus RE, showing that RE is superior for automatic SDB assessment. Results also indicate that, even with the addition of RE signals, the detection rate for hypopnea events is lower, and additional efforts are needed to improve hypopnea detection.

## Figures and Tables

**Figure 1 diagnostics-13-02146-f001:**
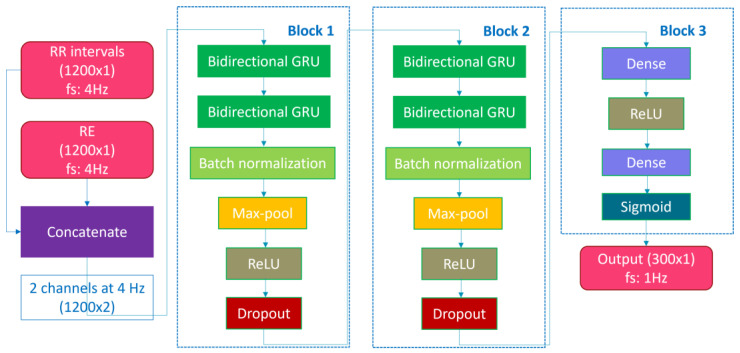
Deep learning model architecture.

**Figure 2 diagnostics-13-02146-f002:**
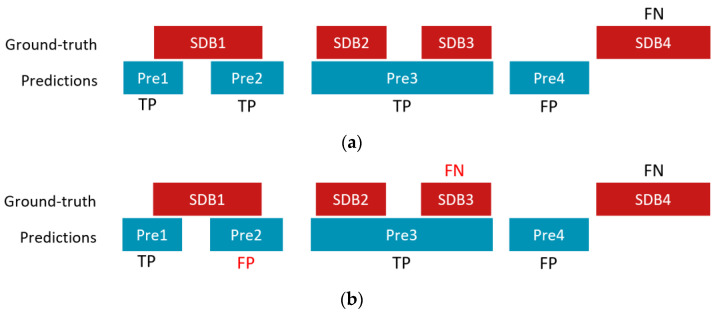
An illustrative example of a hypothetical scenario of counting TP, FP, and FN with four ground truth SDB events (SDB1, SDB2, SDB3, and SDB4) and four detected/predicted events (Pre1, Pre2, Pre3, and Pre 4) using (**a**) rules by Olsen et al. [14] and (**b**) rules used in this work. In this example, Pre2 is counted as a TP according to Olsen’s rules, while it is considered as an FP in this study; SDB3 is considered correctly detected according to Olsen’s rules, but it is treated as an FN based on our rules.

**Figure 3 diagnostics-13-02146-f003:**
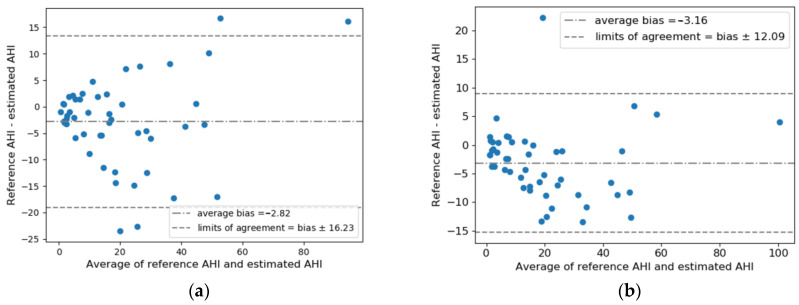
Estimated AHI versus reference AHI values: Bland–Altman plot for (**a**) RR + EDR and (**b**) RR + RE; scatter plot for all test subjects for (**c**) RR + EDR and (**d**) RR + RE; scatter plot for test subjects with AHI < 30 for (**e**) RR + EDR and (**f**) RR + RE. Average bias and limits of agreement and Spearman’s correlation coefficients are indicated in the plots. Note that PSG-based total sleep time was used to calculate the AHI.

**Figure 4 diagnostics-13-02146-f004:**
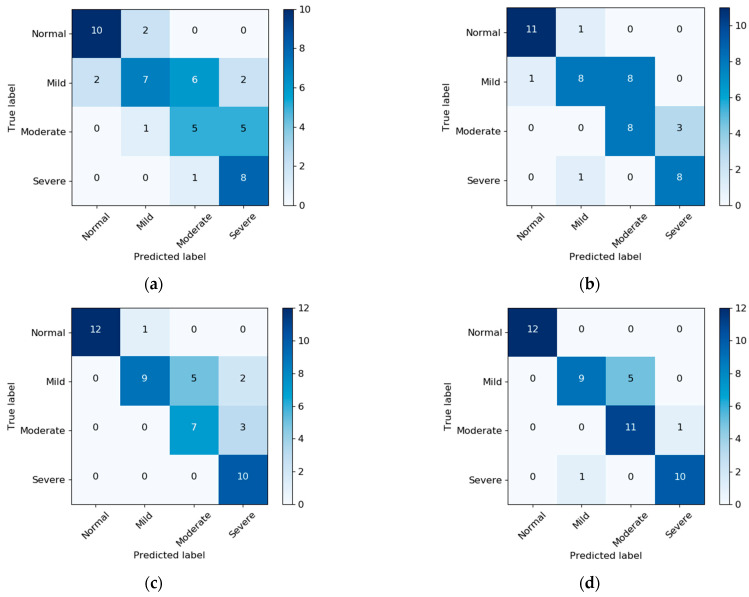
Confusion matrix of SDB severity classification based on AHI: (**a**) RR + EDR without NBL, (**b**) RR + RE without NBL, (**c**) RR + EDR with NBL, and (**d**) RR + RE with NBL. Note that PSG-based sleep information was used for AHI estimation.

**Table 1 diagnostics-13-02146-t001:** General deep learning architecture used for training a SDB event detection model (same as Olsen’s algorithm [14]).

	Layer Type	Kernel Size	Stride	Output Dimension–Explicit
	Input layer	-	-	(B, 1200, 2)
Block1	Bidirectional GRU	128	1	(B, 1200, 128 × 2)
	Bidirectional GRU	128	1	(B, 1200, 128 × 2)
	Batch normalization	-	-	(B, 600, 256)
	Max-pool	-	2	(B, 600, 256)
	ReLU	-	-	(B, 600, 256)
	Dropout (50%)	-	-	(B, 600, 256)
Block2	Bidirectional GRU	128	1	(B, 600, 128 × 2)
	Bidirectional GRU	128	1	(B, 600, 128 × 2)
	Batch normalization	-	-	(B, 600, 256)
	Max-pool	-	2	(B, 300, 256)
	ReLU	-	-	(B, 300, 256)
	Dropout (50%)	-	-	(B, 300, 256)
Block3	Dense	512	1	(B, 300, 512)
	ReLU	-	-	(B, 300, 512)
	Dense	1		(B, 300, 1)
	Sigmoid	-	1	(B, 300, 1)

**Table 2 diagnostics-13-02146-t002:** Participant demographics, and characteristics of training, validation, and testing sets.

	Training	Validation	Testing
Male/Female	63/51	23/12	35/14
Age (years)	49.7 ± 14.9 (range: 18–86)	51.5 ± 13.4 (range: 24–73)	50.0 ± 15.5 (range: 21–79)
BMI (kg/m^2^)	27.0 ± 4.5 (range: 20.0–40.8)	27.1 ± 4.6 (range: 18.6–45.2)	27.6 ± 5.4 (range: 19.9–43.4)
AHI (events/hour)	17.5 ± 17.0 (range: 0–70.6)	19.2 ± 21.0 (range: 0.2–108.4)	18.2 ± 19.4 (range: 0–102.6)
AHI < 5 (count)	28	8	12
5 < AHI < 15 (count)	38	12	17
15 < AHI < 30 (count)	27	8	11
AHI > 30 (count)	21	7	9
Number of hypopneas (count)	92.6 ± 88.2 (range: 0–475, total: 10,560)	89.8 ± 73.8 (range: 1–251, total: 3143)	92.8 ± 81.1 (range: 0–303, total: 4546)
Number of obstructive apneas (count)	9.6 ± 21.4 (range: 0–168, total: 1094)	14.6 ± 53.6 (range: 0–315, total: 511)	11.0 ± 26.2 (range: 0–160, total: 537)
Number of central apneas (count)	5.7 ± 15.0 (range: 0–99, total: 652)	14.0 ± 42.7 (range: 0–249, total: 489)	9.5 ± 26.2 (range: 0–158, total: 464)
Number of mixed apneas (count)	3.4 ± 12.2 (range: 0–104, total: 385)	6.9 ± 27.0 (range: 0–160, total: 242)	9.2 ± 37.8 (range: 0–254, total: 453)
Number of segments	19,760	6043	8425
Number of label 0 (seconds)	5,445,189	1,643,358	2,302,014
Number of label 1 (seconds)	482,811	169,542	225,486

Notes: 1.Statistics present subject mean ± standard deviation, and between parenthesis, range. 2. For different ranges of AHI, the table indicates the number of participants in the corresponding AHI range. 3. The total number of events (indicated between parenthesis), counts the aggregated totals, summed across all subjects. 4. Number of label 1 and label 0 are calculated after segmentation.

**Table 3 diagnostics-13-02146-t003:** SDB event detection results using models trained from different signals (RR + EDR and RR + RE) with and without PSG sleep information.

		Sensitivity (%)	Precision (%)	F1 Score
RR + EDR	mean ± SD	53.2 ± 25.6	41.6 ± 25.0	0.437 ± 0.234
	pooled	65.5	56.5	0.607
RR + RE	mean ± SD	62.6 ± 26.7	50.4 ± 23.6	0.529 ± 0.241
	pooled	77.4	65.2	0.708

**Table 4 diagnostics-13-02146-t004:** Detection rate or sensitivity of different SDB types (results were pooled over all subjects).

SDB Event Type	Total Number of Events	Detection Rate (%)	
		RR + EDR	RR + RE
Hypopnea	4546	59.9	72.4
Obstructive apnea	537	81.9	94.4
Central apnea	464	78.7	90.5
Mixed apnea	453	89.0	94.7

**Table 5 diagnostics-13-02146-t005:** Overall accuracy and Cohen’ Kappa of SDB severity classification based on AHI with and without NBL.

	RR + EDR	RR + RE
Without NBL		
Accuracy	0.612	0.714
Cohen’s Kappa	0.49	0.62
With NBL		
Accuracy	0.776	0.857
Cohen’s Kappa	0.70	0.81

## Data Availability

Currently, the dataset supporting the conclusions of this article is not publicly available. Data can be made available in collaboration with researchers, depending on reasonable request and respecting privacy regulations.

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
