# Peer review of "The Use of Respiratory Effort Improves an ECG-Based Deep Learning Algorithm to Assess Sleep-Disordered Breathing"

_diagnostics, 2023, doi:10.3390/diagnostics13132146_

Round 1
Reviewer 1 Report
The authors of this manuscript evaluated a previously published method for detection of sleep-disordered breathing (SDB) and compared two models for detection of SDB including the model employing RR+RE and the model utilizing RR+EDR and demonstrating the superiority of the RR+RE model. This study is interesting and important and the manuscript is well written. My only concern is that the “superior performance” RR+RE model still shows a relatively low average sensitivity of 63% and precision of 50%. How useful and beneficial is this method in comparison with other methods mentioned in the Introduction Section? For example, you mentioned that Zarei and Asl’s method reached a sensitivity of 92%, a specificity of 94%, and an accuracy of 93%.”
Also, Page 53. Please correct this sentence “...functions of both heart rate variability (HRV or RR interval)…”
Author Response
Response to Reviewer 1 Comments
The authors of this manuscript evaluated a previously published method for detection of sleep-disordered breathing (SDB) and compared two models for detection of SDB including the model employing RR+RE and the model utilizing RR+EDR and demonstrating the superiority of the RR+RE model. This study is interesting and important and the manuscript is well written.
We genuinely appreciate your positive feedback and encouragement.
Point 1: My only concern is that the “superior performance” RR+RE model still shows a relatively low average sensitivity of 63% and precision of 50%. How useful and beneficial is this method in comparison with other methods mentioned in the Introduction Section? For example, you mentioned that Zarei and Asl’s method reached a sensitivity of 92%, a specificity of 94%, and an accuracy of 93%.”
Response 1: Thank you for addressing this, as it pertains to an important point of this study. We have now clarified this by adding a paragraph to the discussion section on page 11 as shown below.
“In comparison to the studies [10–12] described in the introduction, which demonstrated highly promising performance, the results of this study appear slightly lower. Notably, these studies made use of the Apnea-ECG dataset, which is known to yield superior performance compared to other datasets. This assertion is supported by the findings of Papini et al.[13], who observed a significant decline in performance (sensitivity<55%, false detection rate>40%) when applying algorithms trained on the Apnea-ECG dataset (with sensitivity>85% and false detection rate<20%) to alternative databases encompassing a broader spectrum of apneic events and other sleep disorders. Similarly, Yang et al. [11] reported a decrease in all evaluation metrics when validating their method on a different dataset (from accuracy 90.3%, sensitivity 87.6% and specificity 91.9%, to 75.1%, 61.1%, and 80.8% respectively). This aspect was one of the considerations that led to our selection of the study conducted by Olsen et al. [14], as it relied on a dataset comprising large population cohorts. In contrast to the Apnea-ECG dataset, the dataset utilized in the present study was collected in real-world conditions in principle providing a more representative sample. Additionally, it should be noted that while the aforementioned studies focused on segment classification with or without SDB events, our results are based on the detection of actual SDB events; this stricter criterion also contributes to the observed differences in performance.”
Point 2: Also, Page 53. Please correct this sentence “...functions of both heart rate variability (HRV or RR interval)…”
Response 2: The sentence has been changed to:
“For instance, Tripathy [10] used a kernel extreme learning machine classifier on features extracted from the intrinsic band functions of both heart rate variability (HRV or RR interval) and EDR signals and obtained a sensitivity and a specificity for the detection of sleep apnea epochs of 78.02% and 74.64%, respectively.”

Reviewer 2 Report
Part of literature is very old. In my opinion methodology must be more wide described not only using link to literature. Also data set is quite old, from this time are more accurate techniques for data collection and the main question how described methodology works with other data sets
Author Response
Response to Reviewer 2 Comments
Point 1: Part of literature is very old.
Response 1: You have raised a valid point regarding the age of certain literature references. However, we consider these papers as “classical works” that provide an important foundation for our study. Thus, we believe it is essential to include them in our reference list to establish the context and build upon their findings. Moreover, it is worth noting that our primary comparison is made with the study conducted by Olsen et al. [14], which was published in 2020.
Point 2: In my opinion methodology must be more wide described not only using link to literature.
Response 2: Thank you for the invitation to expand on the description of our methods; this is indeed relevant and was also asked for by another reviewer. We have now added more details to section 2.2 (description of the preprocessing methods and procedures to combine different channels) and 2.3 (the training process). In addition, we added a block diagram (Figure 1) to complete the detailed model architecture.
Point 3: Also data set is quite old, from this time are more accurate techniques for data collection and the main question how described methodology works with other data sets
Response 3: Our dataset was collected in 2017 according to the current guidelines from the American Academy of Sleep Medicine for clinical polysomnography. The collection of ECG and RE signals used state-of-the-art technology as used in current clinical practice.
Regarding the main question, it is important to note that a previous version of this algorithm was indeed tested in the original paper by Olsen et al. [14] on different datasets, including two large population cohorts encompassing a total of 9704 recordings. Building upon this previous work, the objective of our study was to extend the evaluation of the method by applying it to a different dataset. Additionally, we sought to assess whether the algorithm's performance could be enhanced by substituting EDR with RE.

Reviewer 3 Report
The authors of this study investigate the application of an ECG-based deep learning algorithm for assessing sleep disordered breathing (SDB). They compare the performance of this algorithm when using electrocardiographic-derived respiration (EDR) and thoracic respiratory effort, finding that the latter provides superior results. However there are few points that should be addressed :
1. It would be useful to have more details about the data preprocessing steps used in the deep learning algorithm for ECG signals.
2. The authors could provide further information about the deep learning model architecture and why it was chosen for this specific problem.
3. How was the algorithm trained to distinguish between normal and abnormal breathing patterns in ECG signals? More details about the training data would be useful.
4. It would be beneficial to understand how the system deals with signal noise and artefacts common in ECG data.
5. How was the performance of the ECG-based deep learning algorithm compared with the standard polysomnography test for sleep apnea detection?
6. What is the method to combine the ECG signals with the respiratory effort signals? It is unclear how these two modalities are fused in the algorithm.
7. What were the sizes of the hypopnea events that the system struggled to detect? Was the detection problem more pronounced with milder events?
8. Were there any specific patient characteristics that influenced the performance of the algorithm, such as age, BMI, or the presence of cardiovascular diseases?
9. How well did the algorithm perform in different sleep stages and body positions?
10. Was the RIP-belt measurement system used in this study commercially available, or was it a custom setup?
11. More information on how the authors dealt with overfitting and underfitting during the training of the deep learning model would be beneficial.
12. How does the computational cost of the ECG-based deep learning algorithm compare with traditional methods for sleep-disordered breathing assessment?
13. The authors should provide more explanation about the validation process of the algorithm.
14. Would integrating other physiological signals like SpO2 (blood oxygen saturation) improve the performance of the model?
15. What are the limitations and potential improvements of the current version of the deep learning algorithm?
16. Given the superiority of the RIP-belt method, how does the team envisage their ECG-based method being used in real-world applications? Are there scenarios where it could be more advantageous?
Author Response
Response to Reviewer 3 Comments
The authors of this study investigate the application of an ECG-based deep learning algorithm for assessing sleep disordered breathing (SDB). They compare the performance of this algorithm when using electrocardiographic-derived respiration (EDR) and thoracic respiratory effort, finding that the latter provides superior results. However there are few points that should be addressed:
Point 1: It would be useful to have more details about the data preprocessing steps used in the deep learning algorithm for ECG signals.
Response 1: Thank you for your valuable and concrete suggestions to further improve our manuscript.
We acknowledge the importance of providing detailed methodology of preprocessing steps. This was also requested by another reviewer. We have now changed the 1st paragraph of section 2.2 by adding more details as below:
“R-peaks were first detected from the ECG signals using a previously developed algorithm which is based on nonlinear transformation and a simple peak-finding strategy [26]. Subsequently, a post-processing algorithm based on the intersection of tangents fitted to the slopes of the R-waves was applied to achieve precise localization of the QRS complexes and to eliminate artifacts [27]. To address the presence of ectopic beats, the algorithm proposed by Mateo and Laguna [28] was employed, which identifies abnormal beats by imposing restrictions on the allowable acceleration or deceleration of the heart rate. The resulting RR intervals were linearly interpolated and resampled at 4 Hz to obtain an RR time series. Periods containing artifacts and ectopic beats were marked with a value of 0 to exclude them from further analysis. The algorithm considered these as missing values and learned how to handle them without the need of using a masking layer.”
Point 2: The authors could provide further information about the deep learning model architecture and why it was chosen for this specific problem.
Response 2: We appreciate the significance of providing clear and comprehensive details regarding the model architecture. To address this, we have now incorporated a block diagram (Figure 1) in our paper, which serves to further elucidate the architecture employed.
The reasons why this method was chosen are introduced in the 4th paragraph in the introduction section on page 2: 1. Their model shows promising results on large population cohorts. 2. EDR can be easily replaced by RE to allow for a direct comparison. 3. RNN can automatically extract features from the signals.
We have now highlighted the first reason again in the 4th paragraph of the discussion section on page 11 by adding
“This aspect was one of the considerations that led to our selection of the study conducted by Olsen et al. [14], as it relied on a dataset comprising large population cohorts.”.
Point 3: How was the algorithm trained to distinguish between normal and abnormal breathing patterns in ECG signals? More details about the training data would be useful.
Response 3: For the training process, we have now added the following to the 3rd paragraph of section 2.3 on page 5.:
“During the training phase, the model was trained using input segments to generate their corresponding target vectors. The patterns for distinguishing between normal and abnormal breathing are learned by the model. Similar to the training process of Olsen’s algorithm, no additional techniques were employed.”
In addition, we provide more details regarding the training validation and testing of the in an additional table (Table 2).
Point 4: It would be beneficial to understand how the system deals with signal noise and artefacts common in ECG data.
Response 4: To make the method more clear, we now have changed the sentence in the 1st paragraph of section 2.2 to
“Periods containing artifacts and ectopic beats were marked with a value of 0 to exclude them from further analysis. The algorithm considered these as missing values and learned how to handle them without the need of using a masking layer.”
The RE signal was down sampled to 4 Hz, thus high frequency noise above 2 Hz was removed. In addition, we applied a high-pass filter with a cut-off frequency of 0.05 Hz to remove low frequency noise. We clarified this in 2nd paragraph of section 2.2 by adding:
“During the resampling step, high frequency (> 2 Hz) noise was also removed.”
Point 5: How was the performance of the ECG-based deep learning algorithm compared with the standard polysomnography test for sleep apnea detection?
Response 5: If we are understanding correctly, you are asking for a comparison with the performance of OSA diagnostic tests, e.g., inter-rater agreement for AHI scoring based on PSG. To provide an insight on this, we calculated the intraclass correlation coefficient (ICC) [33,34] using two-way random-effects model between AHIest and AHIref. In addition, we compared it with the ICC in literature by adding the paragraph below to the discussion section on page 12.
“The AHI estimation performance demonstrates great promise despite the inter-subject variability, with an achieved ICC value of 0.939. This level of agreement is comparable to the AHI agreement observed across 10 scorers in PSGs (ICC 0.984; 95% confidence interval: 0.977-0.990) [38] and the AHI agreement across 9 scorers in home sleep testing data (ICC 0.96; 95% confidence interval: 0.93-0.99) [34]. These results highlight the strong concordance between our AHI estimation and established scoring methods, indicating the reliability of the approach.”
In addition, changes are also made in the 5th paragraph in section 2.3 on page 7:
“and the intraclass correlation coefficient (ICC) [33,34] using two-way random-effects model for measuring the agreement between two scorings are provided”
and in the 1st paragraph of section 3.2 on page 9:
“Consistently, RR+RE exhibited a higher ICC of 0.939 (95% confidence interval: 0.87-0.97) between AHIest and AHIref compared to RR+EDR, which had an ICC of 0.894 (95% confidence interval: 0.81-0.94).”
Point 6: What is the method to combine the ECG signals with the respiratory effort signals? It is unclear how these two modalities are fused in the algorithm.
Response 6: Thank you for pointing out this. We now augmented the 2nd paragraph of section 2.2. to make this more clear.
“Each combination was achieved by directly concatenating segments from the two different signals. Later, the network was trained to autonomously learn the underlying relationship between the two inputs.”
Point 7: What were the sizes of the hypopnea events that the system struggled to detect? Was the detection problem more pronounced with milder events?
Response 7: Indeed, a larger proportion of missed events of all types are of shorter duration, suggesting that milder events (at least as far as they have a shorter duration) are more difficult to detect with the proposed algorithm.
We have now included a boxplot in the supplementary material, depicting the event length distribution of various types of SDB events in the testing set. Additionally, two distinct boxplots have been added to the supplementary material: one illustrates the event length distribution of detected and missed SDB events using RR+EDR, while the other displays the event length distribution of detected and missed SDB events using RR+RE.
Point 8: Were there any specific patient characteristics that influenced the performance of the algorithm, such as age, BMI, the influence from the age, BMI, and comorbidity?
Response 8: Thank you for the suggestion. We have now added the Spearman's correlation coefficient (R) between the error of AHI (|AHIref - AHIest|) and age, BMI to check whether these influenced the performance.
Results were added to section 3.5 on page 11:
“The correlation coefficient was R=0.256 (p<0.1) between age and AHI error and R=0.342 (p<0.05) between BMI and AHI error.”
We added the following to 8th paragraph of the discussion section on page 12.
“Based on our findings, we observed a correlation between AHI error and age as well as BMI. Higher age and BMI were associated with an increased AHI error. However, it is important to note that this correlation might be influenced by AHI severity, as age and BMI are known to be correlated with AHI severity. In Figure 3, we can observe that higher AHIs are associated with a higher AHI error. Nonetheless, further investigations are warranted to establish and validate the influence of age and BMI, on AHI error.”
For the influence from comorbidity, there is too limited information on somatic comorbidities in our dataset, and thus we cannot address this question reliably. The datasets used by Olsen et al. [14] were more appropriate in this aspect and they reported no significant influence of the presence of cardiovascular disease for the model. Additionally, we believe this is an interesting future research direction and have added this to the further work in the last paragraph of the discussion section on page 13.:
“The method should be further validated with larger datasets that contain a wider spectrum of apneic events, additional sleep disorders and additional somatic comorbidities.”
Point 9: How well did the algorithm perform in different sleep stages and body positions?
Response 9: To explore the performance in different sleep stages and body positions, we calculated the event detection rate (sensitivity) in different sleep stages (Supplementary, Table S1) and body positions (Supplementary, Table S2). Furthermore, we added the following to the 8th paragraph in the discussion section on page 12:
“Regarding the lower sensitivity observed during the N3 stage, it is likely attributed to the limited number of events observed in N3, which poses difficulties for our model to effectively learn from them, resulting in a lower sensitivity. In terms of body position, although the percentages of different types of SDB events vary, we found consistent sensitivities in the right, supine, and left positions. The lower sensitivity observed in the prone position may be attributed to the same reason which causes the lower sensitivity in N3. It is worth emphasizing that further research is required to dive deeper into these relationships, confirm the findings, and provide a comprehensive understanding of the factors influencing AHI error and sensitivity.”
Changes are also made in section 2.3 by adding “Additionally, we analyzed the detection rate (sensitivity) in different sleep stages and body positions to explore their association with the event detection performance.”
and in section 3.5 by adding “Regarding the impact of sleep stage on sensitivity (Supplementary Table S1), we observed that sensitivity is lower in N3 (0.448) compared to REM (0.757), N1 (0.784), and N2 (0.772). In terms of body position (Supplementary Table S2), sensitivity was found to be lower in the prone position (0.634) compared to the right (0.738), supine (0.804), and left (0.781) position.”
Point 10: Was the RIP-belt measurement system used in this study commercially available, or was it a custom setup?
Response 10: The RIP-belt is available from Sleepsense (USA). We have now added this information to the section 2.1:
“Among all signals from PSG, only ECG and RE signals are used in this study. ECG signal is measured by electrodes from Kendall (Ashbourne, Ireland) and RE signal is measured by RIP-belts from Sleepsense (Elgin, USA).”
Point 11: More information on how the authors dealt with overfitting and underfitting during the training of the deep learning model would be beneficial.
Response 11: As reported in section 2.3, we have indeed taken measures to counteract overfitting, including the use of a dropout layer, kernel regularization, sample weighting (to compensate for class imbalance) and the use of a separate validation set to enable early stopping (i.e., stopping training when evaluation loss stops decreasing after a number of iterations). Regarding underfitting, we did not take any measures since both the training, validation and hold-out testing performance achieved high accuracy on the classification task at hand. To clarify this, we have added the following to the 2nd paragraph of section 2.3 on page 4:
“(...) we included kernel regularization to counteract overfitting and employed sample weighting (label 0: 1, label 1: 10) to account for the imbalance in our dataset. The training was terminated when the evaluation loss did not improve for more than 20 training epochs, to further avoid overfitting. No underfitting was observed, as both the training and the validation performance achieved high accuracy on the classification task at hand.”
Point 12: How does the computational cost of the ECG-based deep learning algorithm compare with traditional methods for sleep-disordered breathing assessment?
Response 12: Directly comparing the computational cost of different methods is outside the scope of this study. Our study was to provide an option for sleep-disordered breathing assessment based on alternative signals which should be easier to obtain, for example at home, than polysomnography. We do agree that understanding the computational costs of our method is an essential step before this technology can be actually eventually deployed in real-world applications. We added a sentence to the discussion suggesting this for future work.
“(...) it would be relevant to investigate whether the computational requirements needed for this method would be adequate for implementation in modern architectures, embedded in the monitoring devices, or as part of cloud-based services.”
Point 13: The authors should provide more explanation about the validation process of the algorithm.
Response 13: The validation set was only used for early stopping and selecting the best threshold. To make the process clear, we have added some detail to the 3rd paragraph in section 2.3 on page 6.:
“Concretely, we applied a threshold from 0.004 to 0.996 with a step of 0.004 on the SDB event detection results from the validation set and computed the corresponding F1 scores. The threshold with the maximum F1 score was selected to compute the results for the testing set.”
Point 14: Would integrating other physiological signals like SpO2 (blood oxygen saturation) improve the performance of the model?
Response 14: Thank you for suggesting this. SpO2 measurements play an important role in detecting SDB events as well as assessing severity/impact. Several studies have utilized SpO2 for SDB detection, yielding promising performance. Thus, adding SpO2 may improve the performance. However, here we focused on RR and RE signals due to their compatibility with user-friendly and unobtrusive sensors such as wearable devices and bed sensors. In contrast, measuring SpO2 reliably with unobtrusive sensors remains challenging. We now added a note on this to the 7th paragraph of the discussion section on page 12. “On the other hand, including other signals like blood oxygen saturation and airflow, which obviously hold potential to offer valuable additional information, but remain challenging to obtain reliably when utilizing unobtrusive methods.”
Point 15: What are the limitations and potential improvements of the current version of the deep learning algorithm?
Response 15: The limitations of the method are addressed in the 9th and 10th paragraphs on Page 13. To clarify, we changed the first sentence to “It is important to note that, the detection rate for hypopneas was lower compared to other SDB types (see Table 4), consistent with the findings by Olsen et al. [14], which is one limitation of this method.”
We agree that potential improvements are worth exploring, but fall outside the scope of this study Therefore, we have added this to the future work in the last paragraph of the discussion section on page 13 : “Additionally, there is potential in studying and developing methods to improve hypopnea detection rates, for example exploring improvements of the deep learning algorithm by adding attention mechanism or transformers.”
Point 16: Given the superiority of the RIP-belt method, how does the team envisage their ECG-based method being used in real-world applications? Are there scenarios where it could be more advantageous?
Response 16: Thank you for this relevant question. The present study aimed at investigating whether cardiorespiratory inputs could be used to evaluate SDB severity without the use of airflow, or SpO2. We chose two modalities that can be, in theory, be accurately estimated from wearable, or “nearable” sensors, and thus, are well suited for deployment e.g. at home, possibly for prolonged periods of time. That means that although the system could be actually deployed as is on monitoring setups where ECG (and in the case of our RE approach, also RIP belts) is available, the larger benefit will probably come from replacing these with less obtrusive, more practical sensors for real-world applications.
Noting that the “pure” ECG-based method (which uses EDR) has the apparent advantage of requiring a single sensing modality (ECG), and although at first sight the RIP-based approach would suggest the need for a second sensor, the fact is that the signals measured by RIP can be obtained from alternative sensors, such bed sensors, chest-worn accelerometers, or wrist-worn PPG sensor devices. Interestingly, these (all arguably more comfortable and less obtrusive than ECG) can also be used to obtain the cardiac information (interbeat intervals) required by our method, meaning that the better performing model (in our case trained with RIP) could be used without an increase in the complexity of the monitoring system, and without a decrease in patient comfort. Obviously, the performance of the algorithm would need to be verified with these surrogate measures of respiratory effort, since they are similar, but not necessarily equivalent, to RIP-belts.
We now address this point in more detail in the discussion:
“Although the use of an additional modality next to ECG might sound less appealing since it represents an increase in the number of sensors, it is relevant to remark that this is amongst the more easily acquirable, with unobtrusive methods. In fact, recent developments in physiological sensing suggest that both the RR and the RE time series used as input can both be accurately measured from bed sensors [19,22], chest-worn accelerometers [20,23,39], or even wrist-worn PPG sensors [21,24]. In contrast, EDR is ECG-specific and likely not easily obtainable with other sensing modalities.”
And further in the discussion, we added a sentence with the explicit mention of future work required:
“Finally, to enable actual real-world applications, two important elements should be investigated. First, there is clearly sufficient information in cardiac and respiratory activity measured with ECG and RIP-belts to accurately assess SDB severity. However, it is necessary to evaluate whether surrogate measures of cardiac and respiratory effort obtained with other sensors yield comparable performance. (...)”

Reviewer 4 Report
The authors present a methodology for SDB detection and compared the performance between models using EDR versus RE, showing that RE is superior for automatic SDB assessment. The paper is well written, and the authors refer to all the proper references which provide a solid basis of the state-of-the-art and the innovation introduced by this paper.
Before resubmitting, I would suggest the authors to address the following points:
1) It seems fro Figure 1, that the algorithm may detect a higher number of SDB events. Have the authors checked for each patient the actual SDB overlapping pattern? Is there a correlation between AHI in the case where SDB_detected > SDB_actual?
2) What about class imbalancing? Is there inter-subject variability in the 198 patients the authors used?
The English language of the paper is OK.
Author Response
Response to Reviewer 4 Comments
The authors present a methodology for SDB detection and compared the performance between models using EDR versus RE, showing that RE is superior for automatic SDB assessment. The paper is well written, and the authors refer to all the proper references which provide a solid basis of the state-of-the-art and the innovation introduced by this paper.
We appreciate the positive reception of our paper.
Before resubmitting, I would suggest the authors to address the following points:
Point 1: It seems from Figure 1, that the algorithm may detect a higher number of SDB events. Have the authors checked for each patient the actual SDB overlapping pattern? Is there a correlation between AHI in the case where SDB_detected > SDB_actual?
Response 1: We agree that it may seem from Figure 1 that the algorithm might detect a large number of SDB events for each actual SDB event, which would warrant a proper labelling of falsely detected positives in that case. To evaluate whether that was the case, we compared the number of additional false positives obtained with the stricter rule of Figure 1 b) versus the rule of Figure 1 a) where multiple overlapping detections were all counted as true positives. We found that for our RR+RE algorithm there were only a total of 43 additional false positives scored with the stricter rule suggesting that the detection of multiple SDB events for each actual SDB event was not an actual issue with our algorithm.
Regarding your second question, we evaluated whether there was a correlation between the AHI and the difference between the number of detected SDB events and the number of actual SDB events (i.e., whether the error in the number of detected events was correlated with the severity of the condition). We found a non-significant correlation of 0.21 (p=0.14) suggesting that the two are not associated. We added a sentence indicating this to the end of section 3.1:
“Furthermore, for the model with RR+RE, we found a non-significant Spearman’s correlation of 0.212 (p=0.14) between the error in the number of detected events (i.e., true positives plus false positives) minus the number of actual events according to the ground-truth, and the reference AHI.”
Point 2: What about class imbalancing? Is there inter-subject variability in the 198 patients the authors used?
Response 2: Thank you for pointing out this unclear part. There is indeed class imbalance: the number of seconds with label 0 (normal) is much bigger than that of label 1 (SDB events). There is also inter-subject variability with a wide range of AHIs and variations in the distribution of event types scored across different subjects. To provide further clarity, we have now included Table 2, which illustrates the imbalance and variability present in the training, validation, and testing sets. Importantly, despite the inter-subject variability, our method consistently achieves accurate AHI estimation for the subjects. We have now added this to 5th paragraph in the discussion section on page 12:
“The AHI estimation performance demonstrates great promise despite the inter-subject variability”.

Round 2
Reviewer 3 Report
Authors corrected the all the comments
Reviewer 4 Report
The authors addressed the revisions successfully.
English is ok.